# Preparation and Hydrogen Storage Characteristics of Surfactant-Modified Graphene

**DOI:** 10.3390/polym10111220

**Published:** 2018-11-02

**Authors:** Tao Xu, Jiayu Chen, Wenhui Yuan, Baoqing Li, Li Li, Huijun Wu, Xiaoqing Zhou

**Affiliations:** 1Academy of Building Energy Efficiency, School of Civil Engineering, Guangzhou University, Guangzhou 510006, China; xutao9@mail.sysu.edu.cn (T.X.); wuhuijun@tsinghua.org.cn (H.W.); zhou_xiaoqing03@163.com (X.Z.); 2Department of Architecture and Civil Engineering, City University of Hong Kong, Tat Chee Avenue, Kowloon, Hong Kong, China; jiaychen@cityu.edu.hk; 3Key Laboratory of Enhanced Heat Transfer and Energy Conservation, The Ministry of Education, School of Chemistry and Chemical Engineering, South China University of Technology, Guangzhou 510640, China; lbqscut@163.com; 4School of Environment Energy, South China University of Technology, Guangzhou 510006, China; lli@scut.edu.cn

**Keywords:** surfactant, graphene, hydrogen storage, adsorption

## Abstract

As the depletion of traditional fossil fuels and environmental pollution become serious problems for human society, researchers are actively looking for renewable energy sources. Since hydrogen energy is considered a clean, efficient, and renewable alternative energy source, it is regarded as the most promising option. In this context, how to store hydrogen safely and efficiently has become the major challenge that hinders the actual application. To fill this gap, this paper proposes to utilize surfactant-modified graphene for hydrogen storage. Through a modified Hummers’ method and ultrasonic stripping, this study proposes to prepare graphene from graphite oxide with NaBH_4_. The surfactant sodium dodecyl benzene sulfonate (SDBS) was used as a dispersant during the reduction process to produce dispersion-stabilized graphene suspensions. Then, to investigate the characteristics of the graphene suspensions, X-ray diffraction (XRD), SEM, TEM, Fourier transform infrared (FT-IR), Raman, XPS, TG, and N_2_ adsorption–desorption tests were conducted. Finally, analytical models for hydrogen adsorption were investigated with Langmuir and Freundlich fittings. The results show that the application of SDBS can effectively reduce the agglomeration among graphene monolayers and increase the specific surface area of graphene, and that the adsorption behavior is consistent with the Freundlich adsorption model, and is a physical process.

## 1. Introduction and Background

Energy and the environment are two major issues that mankind needs to confront. With the depletion of traditional fossil fuels and serious environmental pollution, researchers have come to an agreement on the necessity of alternative clean and renewable sources. Hydrogen energy is recognized as one of the ideal options for its cleanness, efficiency, and renewability [1,2,3,4,5,6]. The energy density obtained from hydrogen is almost three times higher than that of gasoline, and about seven times higher than that of coal in an equal mass [7]. However, hydrogen is unstable, and in order to safely and efficiently store hydrogen is a crucial barrier impeding its wide application. Proper hydrogen storage should involve high energy density, low energy waste, and high reliability. For example, the renewable energy generating systems in buildings rely on hydrogen to store the produced energy for commercial buildings [8,9] and infrastructures [10].

Researchers usually use mass storage density and volume storage density as two major indicators for hydrogen storage. Many researchers have carried out extensive and profound studies on hydrogen storage technologies in past decades [11,12]. One branch of study involves the storage materials, which categorizes hydrogen storage as physical adsorption and chemisorption based on material types. Good hydrogen storage materials should allow rapid storage and a high volumetric and gravimetric density of hydrogen. There are four main types of adsorption materials, including carbon hydrogen storage materials, non-carbon nanotube materials [13], mineral porous materials, and microporous metal–organic frameworks [14]. The carbon hydrogen storage materials are more popular, as they are insensitive to a small amount of gaseous impurities, and can be used repeatedly. In addition, they have low atomic mass, high chemical stability, and rich tubing structures [15,16,17,18,19,20]. Carbon hydrogen storage materials also have subcategories, including super activated carbon, nanostructured carbon materials, and graphene [21]. Graphene is considered as the prospective new carbonaceous hydrogen storage material with the highest potential due to its unique physical and chemical properties and simplicity of preparation. When modified by metals such as Pd, Pt, Ni, Ti, Sc, V, Cu, etc., graphene can further improve its hydrogen storage capacity [22,23,24].

The preparation of graphene has made great breakthroughs in past decades. Recent studies have mainly focused on increasing the specific surface area of graphene materials. Srinivas et al. oxidized graphite with a modified Hummers’ method and reduced hydrated hydrazine to produce graphene [25]. Under a pressure of 1000 kPa and temperatures of −196 °C and 25 °C, the absorbed amount of hydrogen can reach 1.2 wt % and 0.1 wt %, respectively. Ghosh et al. produced graphene through the thermal exfoliation of graphite oxide at high temperature [26]. Their results show that the hydrogen adsorption capacity was about 2.0–3.1 wt % at 10,000 kPa and 65 °C. Cheng et al. reported that the hydrogen adsorption capacity of graphene reaches 0.4 wt % and 0.2 wt % at 100 kPa, −196 °C, 6000 kPa, and 25 °C, respectively [27]. In addition, various graphene preparation studies show that, due to agglomeration, the actual specific surface area of the produced graphene is much smaller than its theoretical value (6.0 wt %). Therefore, this study aims to obtain a stable graphene suspension and prepare high-quality single-layer graphene in order to improve hydrogen adsorption.

Graphene has low hydrophilicity and lipophilicity, which tends to cause agglomeration. Without preprocessing, it is difficult to obtain monolayers and aqueous solutions or organic reagents. The dispersion of graphene may form lumps and agglomerates, which further deteriorate its performance after drying. Therefore, it is necessary to improve the hydrophilicity or lipophilicity of graphene so that the formation of composites and other substances can be reduced. In recent years, researchers found that the surface of graphite oxide contains many active groups, such as carbonyl groups, carboxyl groups, epoxy groups, etc. These groups can react with other substances, making the modification of graphene feasible, and thus leading to an active graphene surface. In general, the modification methods for graphene mainly include surface modification, chemical doping, functionalization of a polymer base, and physical modification. Bourlino et al. utilized amino siloxane and amino acid for its modification, and the modified graphene can remain stable in water [28]. Niyogi et al. selected octadecylamine (ODA) as a modifier to synthesize long-chain alkylated graphene on the surface of graphite oxide [29]. The modified graphene can disperse in tetrahydrofuran and carbon tetrachloride solutions. Ensafi et al. used a spillover mechanism to improve the hydrogen storage capacity of layered double hydroxides/reduced graphene, and the presence of *o*-phenylenediamine and *p*-phenylenediamine improved the kinetics of the hydrogen adsorption [30]. Zhao et al. used the graphene sulfonation method, which can not only improve the graphene, but also maintain the properties of graphene well [31]. This method first uses the sodium borohydride to allow a reduction–oxidation reaction, and then, the product was sulfonated in an ice bath for 2 h. Finally, bisamine was used as a reducing agent for chemical reduction. Liang et al. used charge repulsion to obtain a well-dispersed graphene suspension in water [32]. Niyogi et al. reported that octadecylamine can be used to treat long-chain alkyl modified graphene [29]. Based upon the aforementioned studies, this paper proposes a new method to prepare high-quality graphene for the purpose of hydrogen storage.

## 2. Methodology

### 2.1. Materials and Apparatus

The experiment materials and equipment are summarized in following Table 1 and Table 2.

### 2.2. Preparation of Surfactant Modified Graphene 

#### 2.2.1. Preparation of Graphite Oxide

A modified Hummers’ method was used to prepare graphite oxide. The experimental method is shown in Figure 1. First, a 500-mL reaction flask was assembled in an ice water bath. Then, 5 g of graphite powder and 5 g of sodium nitrate with 200 mL of concentrated sulfuric acid were added in the flask. Under stirring, 25 g of potassium perchlorate and 15 g of permanganate were introduced. The reaction temperature was kept below 20 °C. If the reactions finished, the reaction bottle was removed from the ice bath and stirred on an electromagnetic stirrer for 24 h. After that, 200 mL of deionized water was added slowly, and the temperature was raised to about 98 °C. After stirring for 20 min, an appropriate amount of hydrogen peroxide was added to reduce the remaining oxidizing agent until the solution became bright yellow. The graphite oxide suspension was then centrifuged at a rate of 10,000 rpm and washed successively with 5% HCl solution and deionized water until the pH of the separation solution reached 7.0. The obtained cake was vacuum dried to obtain graphite oxide (GO).

#### 2.2.2. Preparation of Graphene

Figure 2 shows the method of preparation of graphene by sodium dodecyl benzene sulfonate (SDBS). First, the pulverization of GO was carried out via the spray-drying method. Then, 0.25 g/L GO aqueous solution was pulverized on a spray dryer at 200 °C with a flow rate of 300 mL/h, and a gas flow rate of 0.7 m^3^/h. After the pulverization, 300 mg of graphite oxide was dispersed in 60 mL of deionized water to obtain a brownish-yellow suspension. A stable colloidal suspension can be obtained after ultrasonic dispersion for 1 h. Then, the suspension was transferred to a four-necked flask, and 600 mg of sodium borohydride and 50 mg of sodium dodecyl benzene sulfonate (SDBS) were added at 80 °C. After refluxing for 16 h, the mixture was centrifuged and washed with acetone and deionized water until a pH of 7 was reached. The obtained filter cake was vacuum dried and stored for future use. This product was denoted as GS1. The other graphene that was prepared without a dispersing agent was denoted as GS.

#### 2.2.3. Inspection of Physical Structure

An X-ray diffraction (XRD) test was conducted with a D8 polycrystalline powder diffractometer (Bruker, Ettlingen, Germany) to analyze the crystal structure of the prepared graphite oxide and graphene samples. In the test, with Cu Ka radiation, the λ = 1.54 × 10^−10^ m line was used as the ray source with a tube flow of 30 mA (tube pressure is 40 kV, and the scanning range is 5°–60°). The infrared absorption spectrum of the samples was measured with a Vector 33 Fourier Transform Infrared Spectrometer (Bruker, Germany) with a scanning range of 4000–400 cm^−1^. KBr tablets were used for sample preparation. Then, the samples were tested with a Raman spectrometer (LabRAM Aramis, HJY, Paris, France) at room temperature. The excitation line was a 632.8-nm line of an Ar ion laser. One measurement included 20 scans with 20 mW of power. The X-ray photoelectron spectroscopy (Axis Ultra DLD, Kratos, Manchester, UK) was performed with *Al Kα* radiation (15 kV, 10 mA, hv = 1486.6 eV) as the excitation source. The water content and composition of samples were examined by a TG analyzer (SAT449C, NETZSCH, Bavaria, Germany). The test was carried out under an atmosphere of oxygen (gas flow: 30 mL/min) and nitrogen (gas flow: 28 mL/min), and a temperature between 20–800 °C with a heating rate of 5 °C/min. The TG test aims to verify whether the sample was sufficiently dry and the graphite oxide was completely reduced. The morphology and roughness of the sample surface were inspected with an atomic force microscope (Veeco Multimode 3D, Veeco, NJ, USA). For the inspection, 1 mg/L graphene suspensions with water dispersion medium were prepared. The surface morphology and structure of graphene were also inspected with a scanning electron microscope (S-3700N, Hitachi, Tokyo, Japan). Gold films were sprayed on the surface of the sample with 5 min of gold plating time. The ultrastructure of the graphene surface was observed with a transmission electron microscope (Tecnai G2 F30 S-Twin, Philips-FEI, Amsterdam, The Netherlands). The measurement of an N_2_ adsorption–desorption isotherm was studied with a specific surface area analyzer (ASAP 2010, Micromeritics, Atlanta, GA, USA), which had liquid nitrogen as a dispersion medium (77 K). According to the characteristics of the graphene sample N_2_ absorption capacity, the surface area of the sample was calculated based on the Brunauer–Emmett–Teller (BET) equation [33]. The pore size distribution curve was calculated by the Barrett–Joyner–Halenda (BHJ) method with desorption branches [34].

#### 2.2.4. Inspection on the Hydrogen Storage Capacity

High pressure (0–3000 kPa) adsorption tests of hydrogen gas were carried out on a magnetic suspension balance (Rubotherm, Bochum, Germany). The balance contains an automatic gas sampling system, pressure control system, and temperature control system. In the experiment, the purge gas was high-purity helium (99.999%), and the adsorption gas was high-purity hydrogen (99.999%). First, the sample with the 1/3–1/2 sample frame volume was placed in a stainless steel sample box. Then, the pretreatment of samples was carried out in an electric oven at 150 °C under vacuum for 12 h. Thereafter, the adsorption test was conducted with a controlled flow rate of 30 mL/min for both the adsorption gas and the purge gas. The adsorption temperatures were set to 25 °C and 55 °C, and different measurement points were selected within the pressure range of 100–2500 kPa. The adsorption instrument recorded the sample weights at each predefined pressure level. After data processing, the hydrogen adsorption isotherm under different temperatures can be obtained.

## 3. Characterization of Graphite Oxide and Graphene Samples

### 3.1. XRD Characterization

Figure 3 shows the XRD characteristics of graphite (G), graphite oxide (GO), and graphene (GS1/GS). In the figure, the crystal plane diffraction peak of graphite is extremely strong and sharp at 2*θ* = 26.58°, indicating that the graphite sheets are laid neatly through the space. The graphite oxide crystal plane has a diffraction peak at 2*θ* = 10.60°, with a relatively low intensity. This result suggests that the graphite interlayer spacing increases from 0.336 nm to 0.779 nm due to the oxidation process. When the graphite oxide was reduced to graphene, a diffraction peak appeared at 26.08°, and the diffraction peak became broader with decreased intensity. This can be explained by the decreased gap between the graphene sheets and higher disorder level. In Figure 3b, the diffraction peak of graphene GS1 is lower than that of GS, since the activity of dispersant hinders the agglomeration of graphene sheets. As the graphene GS1 that is prepared is mostly monolayered, the distance between layers is much larger than the wavelength of X-rays, which leads to inconspicuous diffraction. This demonstrates that adding dispersant is effective at creating monolayer graphene.

### 3.2. Raman Spectroscopy

Figure 4 shows the Raman spectra of graphite (G), graphite oxide (GO), and graphene (GS1). The G sample has a sharp and strong absorption peak (G-peak) at 1576 cm^−1^ and a weak absorption peak (D-peak) at 1332 cm^−1^, indicating that the atomic structure sp2 hybrid carbon of graphite is very tight. After the graphite was oxidized, the G-peak of the graphite oxide becomes boarder and blue-shifts to 1578 cm^−1^, while the intensity of the D-peak increases. This result indicates that after oxidization, the structural symmetry decreased, the vibration mode increased, part of the sp2 hybrid carbon atoms were converted into sp3 hybrid structures, and C=C double bonds were destroyed. In addition, the intensity ratio of the D-band to the G-band (ID/IG, sp3/sp2 carbon atomic ratio) suggests that the plane length of the sp2 hybridized carbon layer in graphite oxide is shorter than that of graphite. The ID/IG of graphite is 0.052, which is obviously smaller than that of graphite oxide and graphene. Due to the supersonic effect, after the graphite was oxidized, the graphite layers were broken down, and the graphite layer area had shrunk considerably. After graphite oxide is reduced, the D-peak and G-peak were red-shifted. Its intensity ratio (ID/IG = 1.28) is higher than that of the graphite oxide (ID/IG = 1.06), indicating that after the graphite has been fully oxidized and ultrasonically exfoliated, the average size of the sp2 hybridized carbon layers in graphene was larger than that of graphite oxide. At the same time, only a part of the sp3 hybrid carbon atoms were reduced to the sp2 hybrid carbon atom, that is, the structure of the graphite oxide cannot be completely restored to its original graphite structure.

### 3.3. FT-IR Characterization

Figure 5 shows the infrared spectrum of graphite (G), graphite oxide (GO), and graphene (GS1). Before oxidization, due to the graphite six-membered ring having almost no other groups, the peak on the infrared spectrum is relative weak. After oxidization, peaks corresponding to OH (3414 cm^−1^), C=O (1723 cm^−1^), aromatic ring C=C (1627 cm^−1^), and epoxy CO (1233 cm^−1^) appeared on the infrared spectrum of graphite oxide. The OH vibration peak was generated due to the vibration of hydroxyl groups in the water molecules between the graphite layers. The carbonyl, carboxyl, and epoxy groups result from graphite oxidation. The carbonyl peak is very weak, as it can only be formed close to the graphite layer edge. The epoxy and the alkoxy groups exhibit strong peaks, since the graphite oxide contains a large amount of hydroxyl and epoxy groups. After the graphite oxide was reduced, the oxygen-containing functional group peak of the graphene sample significantly weakened or even disappeared, while the aromatic ring C=C (1627 cm^−1^) peak was enhanced. Most of the oxygen-containing functional groups in the graphene were removed, but the crystal structure is different from that of the original graphite.

### 3.4. XPS Characterization

Figure 6 shows the XPS spectra of graphite oxide and surfactant-modified graphene. The XPS test shows that the characteristic peaks, C1s and O1s, appear at 283 eV and 534 eV, respectively. The carbon-to-oxygen atomic ratios of graphite oxide and graphene are 2.11 and 7.06, respectively, indicating that the graphite is fully oxidized. The XPS spectrum of graphite oxide in Figure 6a shows that the sp2 hybridized C–C skeleton characteristic peak and C–O–C epoxy functional group peak appear at 285 eV and 287 eV, respectively, while the weak carboxyl characteristic peak appears at 289 eV, suggesting the existence of oxygen-containing groups. After reduction, the characteristic peaks of the modified graphene, including epoxy and carboxyl groups, are significantly reduced. This result agrees with results of previous infrared tests [35,36].

### 3.5. TG Analysis

Figure 7 shows the TG curves for GO and GS1. When the graphite oxide was heated to 800 °C, the process has two obvious weight loss steps. The mass loss below 150 °C was mainly caused by the volatilization of the water molecules adsorbed by the graphite oxide, and the mass loss between 180–250 °C may be due to the heat generation in the oxygen-containing groups. The decomposition process generates CO, CO_2_, and H_2_O. The GS1 had a slight mass loss below 160 °C due to the small amount of adsorbed water molecules. Between 160–800 °C, the mass loss was significantly reduced compared to curve (a), indicating that the GS1 sample has better thermal stability.

### 3.6. N_2_ Adsorption–Desorption Characterization

Figure 8 show the N_2_ adsorption–desorption isotherms and pore size distributions of graphene samples GS1 and GS at liquid nitrogen temperature, respectively. From the figure, the isotherm shows a type I adsorption curve, and a hysteresis loop appears on the isothermal adsorption–desorption curve at a relative pressure of about 0.4, which results in a narrow gap between graphene layers. The specific surface area of GS1 (about 1206 m^2^∙g^−1^) is much larger than that of GS (605 m^2^∙g^−1^), but smaller than the theoretical surface area of single-layer graphene (2630 m^2^∙g^−1^). This shows that the graphene sheet GS1 has both single-layer graphene and double-layered structures. However, compared with previous studies, the specific surface is still significantly improved [27]. The pore size distribution in Figure 8b also shows that, compared with GS, GS1 has a smaller pore size and a larger number of micropores. This explains why the specific surface area of GS1 is larger than that of GS.

### 3.7. Morphological Representation

Figure 9 shows the SEM (a, b) and TEM (c, d) images of graphene GS1 and GS, respectively. In Figure 9a, the graphene GS1 is composed of a layer of honeycomb monolithic structures with a well-arranged and porous structure, indicating that there is very little agglomeration during the reduction process. In Figure 9b, graphene GS is stacked by layers, indicating significant agglomeration during the reduction process. Due to the dispersant, the dispersion of GS1 is significantly improved. In Figure 9c, the GS1 has an almost transparent, thin silk-like structure with a smooth, regular surface and fewer defects, while also containing fewer graphene layers. In Figure 9d, the surface of GS is uneven, and its thickness is also non-uniform, indicating that GS includes a large number of overlapped and curled single-layer graphene. The results show that adding dispersant can effectively improve the quality of graphene produced through the reduction of graphite oxide.

Figure 10 displays an atomic force microscopy (AFM) image of the GS1. In the figure, the graphene nanosheets are evenly dispersed on a mica sheet substrate. The sample shows a stacked state with uniform colors that can be clearly distinguished. Stankovich et al. [37] have theoretically demonstrated and experimentally shown that the single-layer graphene produced by the liquid oxidation–reduction method has a thickness of about 1.1 nm. According to the AFM analysis, the average thickness of GS1 is about 1.1 nm, which suggests a single graphene layer, while very few double or triple graphene layers also exist in the sample.

### 3.8. Dispersibility of Graphene Aqueous Solution

Figure 11 compares the solubility, dispersion, and stability of graphene suspension (concentration 5 mg/mL) before and after adding 100 mL of dispersant. It can be seen from the figure that most of the unmodified graphene sinks to the bottom, and obvious delamination and instability occur. The modified graphene remains stable in aqueous solution. The dispersant reduces the surface energy between the particles, which improves the hydrophilicity of the graphene. The modified graphene has a good water solubility and dispersibility, and remains stable for more than 60 days.

## 4. Validations of the Hydrogen Storage Capacity

### 4.1. Hydrogen Adsorption

Figure 12 shows the comparison of experimental results of high-pressure hydrogen adsorption on GS1 and GS prepared under different pressure and temperature. In the figure, the adsorption capacity of graphene GS1 is significantly higher than that of GS. At 25 °C and 55 °C, the hydrogen adsorption of graphene GS1 and GS varies under different pressures. At the same temperature, with the increase of pressure, the adsorption amount on graphene gradually increases; at the same pressure level, the adsorption amount decreases with the increase of temperature.

When the hydrogen pressure is relatively low, hydrogen molecules are mainly adsorbed in the micropores. As GS1 has more micropores, its adsorption capacity is better than GS in the early stages. With the increase of pressure, the hydrogen is mainly adsorbed by the mesopores and macrospores, and the adsorption amount increases dramatically. For example, at 25 °C and 2500 kPa, the amount of hydrogen adsorbed by GS1 is as high as 1.7 wt %. The heat of adsorption of hydrogen for graphene at temperatures *T*_1_ and *T*_2_ can be calculated by the Clausius–Clapeyron equation (as below, Equation (1)) [27], which is approximately 18.6 kJ/mol. This value is close to the adsorption heat (15 kJ/mol) of the ideal adsorbent material suggested by the United States Department of Energy [38].

Clausius–Clapeyron equation:(1)Qst=R·ln(PT1PT2)1T1−1T2
where *Q_st_* is the heat of adsorption (kJ/mol), *R* is the gas constant (8.314 kJ/(mol K)), and *P_T_*_1_ and *P_T_*_2_ are the hydrogen pressures (kPa) at temperatures *T*_1_ and *T*_2_, respectively.

### 4.2. Hydrogen Storage Capacity-Fitting

In order to analyze the rule of adsorption properties of graphene, two kinds of hydrogen adsorption models are discussed.

#### 4.2.1. Langmuir Model

Langmuir isothermal adsorption mode (Equation (2)) is the first model to describe the adsorption mechanism vividly, which lays a foundation for the establishment of other adsorption models. The model assumes that the adsorbent surface is uniform and there is no interaction between the adsorbents. Also, the adsorption is considered as only occuring on the external surface of the adsorbent.

(2)$$\frac{p}{V^{a}}=\frac{1}{V_{m}^{a}\cdot b}+\frac{p}{V_{m}^{a}}$$
where $V_{m}^{a}$ is the gas volume during adsorption saturation, $V^{a}$ is the gas volume during adsorption equilibrium, b is the adsorption equilibrium constant, which is a dimensionless parameter, and $V_{m}^{a}$ and b can be estimated with regression slope and intercept, respectively.

#### 4.2.2. Freundlich Adsorption Model

Freundlich isothermal adsorption model (Equation (3)) can be applied to both single-layer adsorption and non-uniform surface adsorption, which can describe the adsorption mechanism of non-uniform surface and is more suitable for low concentration adsorption.

(3)$$\mathrm{lg}V^{a}=\mathrm{lg}K+n\cdot \mathrm{lg}p\;$$
where *V^a^* is the actual amount of adsorption (mL/mg) under pressure, *K* is the reaction rate constant, *n* is the empirical constant, and *p* is the adsorption pressure. A smaller n means a better adsorption ability. In general, when n is between 0.1–0.5, the adsorption is easy to perform; when *n* > 2, adsorption is more difficult. $V^{a}$ can be measured by the experimental method, and then *K* and *n* can be calculated through the above model. Figure 13 shows the results of both fitting models.

The adsorption isotherms of hydrogen on graphene samples were fitted with Langmuir adsorption isotherm equations and Freundlich adsorption isotherm equations, as shown in Figure 13. The fitting functions of the Langmuir adsorption isotherm equation and the Freundlich adsorption isotherm equation agree with the experiment results. The Freundlich adsorption isotherm equation shows a better fitting result. Table 3 lists the fitting parameters and variances for the adsorption isotherms. The correlation coefficients of both the Langmuir adsorption isotherm equation and the Freundlich adsorption isotherm equation are above 0.94.

Form Table 3, at 298 K and 328 K, the correlation coefficient of the Freundlich adsorption isotherm equation is better than that of the Langmuir adsorption isotherm equation. The Freundlich adsorption isotherm equation was used to describe the adsorption isotherms, and it illustrates the existence of the peripheral curl and stacked states of the graphene sample, which is non-monolayer adsorption. The Langmuir adsorption model emphasizes the interaction between the monolayer and the adsorption medium, while the Freundlich adsorption model is a more flexible semi-empirical equation. The Freundlich adsorption model also shows that the adsorption of hydrogen molecules on graphene is a physical process. When hydrogen is adsorbed on graphene at low pressure, it first occurs in the micropores and hollow structure of graphene. With the increase of equilibrium pressure, the adsorption of hydrogen molecules gradually occurs on the outer surface of graphene, which is in line with the physical adsorption characteristics.

## 5. Conclusions

This paper proposed preparing graphite oxide through a liquid phase oxidation method. After ultrasonic peeling, a high-quality graphene sheet can be prepared by adding the dispersant sodium dodecyl benzene sulfonate (SDBS) during the reduction process. The structure, morphology and composition of the prepared samples were inspected by XRD, Raman, Fourier transform infrared (FT-IR), TG, SEM, TEM, and AFM. The hydrogen adsorption properties of graphene then were investigated by conducting hydrogen adsorption experiments. Through characterization analysis and experimental results, this study concludes:

(1) Adding the dispersant sodium dodecylbenzene sulfonate can greatly improve the quality of the prepared graphene sheets. The final prepared product is mainly composed of single-layer graphene with a thickness of 1.1 nm and a small amount of double-layer graphene.

(2) The N_2_ adsorption–desorption experiments show that the graphene sheet involves a large specific surface area with a rich pore structure. The application of dispersant sodium dodecyl benzene sulfonate (SDBS) can effectively reduce the agglomeration among graphene monolayers and increase the increase the specific surface area of graphene (from unmodified graphene 605 m^2^∙g^−1^ to improved 1206 m^2^∙g^−1^).

(3) In the hydrogen adsorption experiments, the adsorption capacity of high-quality graphene at 25 °C and 55 °C with pressure of 2500 kPa reached 1.7 wt % and 1.1 wt %, respectively. The adsorption isotherms of hydrogen on graphene samples were fitted by the Langmuir adsorption isotherm equation and the Freundlich adsorption isotherm equation. The fitting results show that the linearity of the Freundlich adsorption isotherm equation performs better.

(4) The Freundlich fitting model also shows that when hydrogen is adsorbed on graphene at low pressure, it first occurs in the micropore and hollow structures of graphene. With the increase of adsorption pressure, the adsorption of hydrogen molecules gradually occurs on the outer surface of graphene.

## Figures and Tables

**Figure 1 polymers-10-01220-f001:**
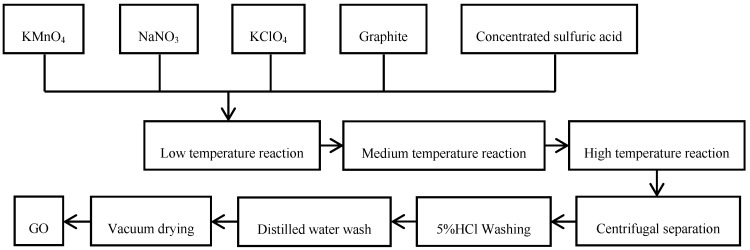
Flow chart of preparation of graphite oxide.

**Figure 2 polymers-10-01220-f002:**
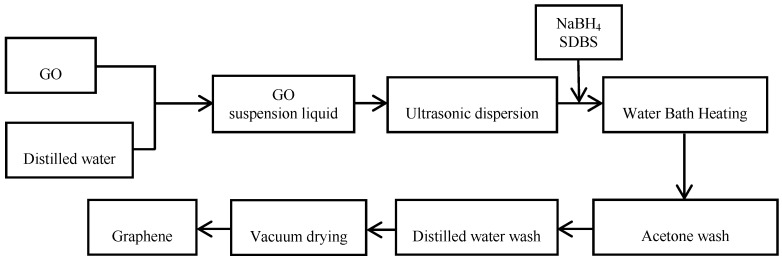
Flow chart of preparation of graphene by SDBS.

**Figure 3 polymers-10-01220-f003:**
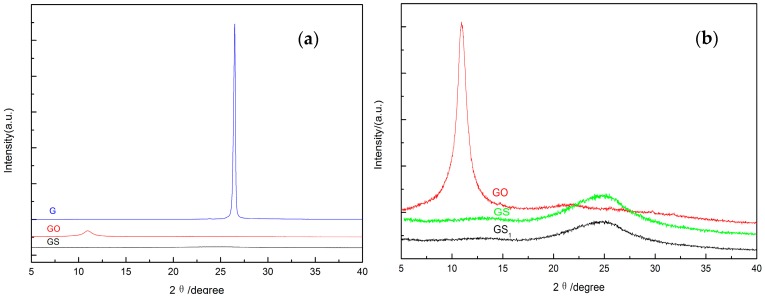
(**a**) X-ray diffraction (XRD) pattern of graphite (G), graphite oxide (GO), and graphene (GS); and (**b**) XRD magnification of GO, graphene (GS1), and GS.

**Figure 4 polymers-10-01220-f004:**
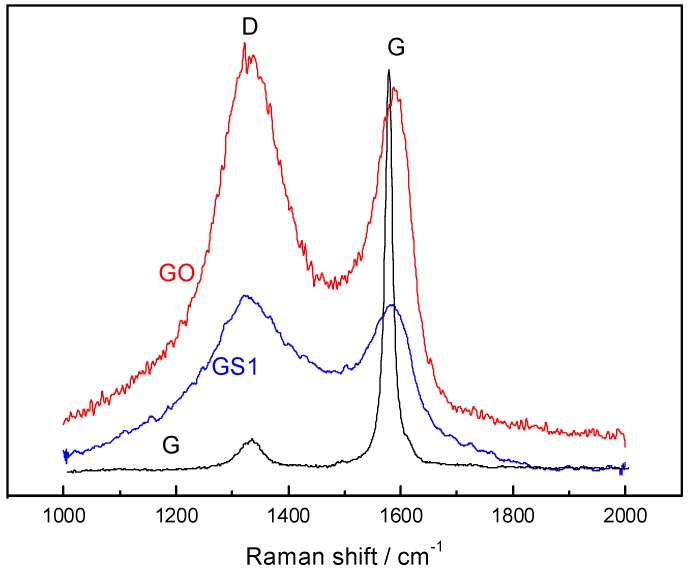
The Raman spectra of G, GO, and GS1.

**Figure 5 polymers-10-01220-f005:**
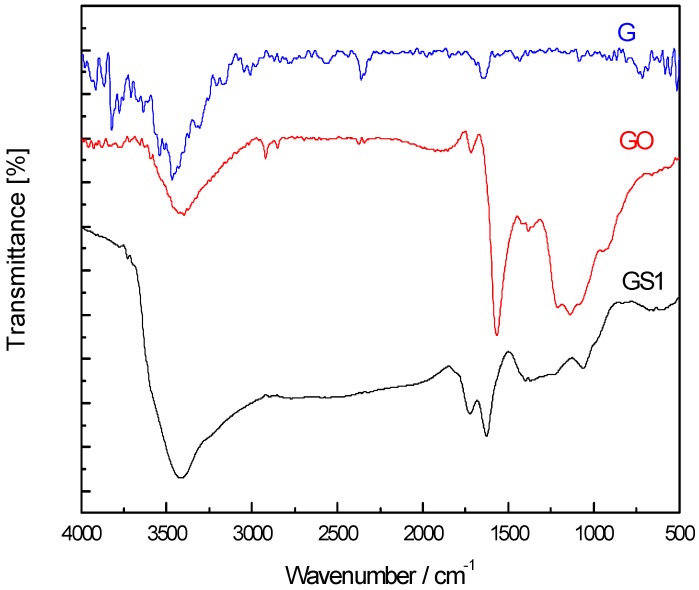
Fourier transform infrared (FT-IR) spectra for graphite, graphite oxide, and graphene.

**Figure 6 polymers-10-01220-f006:**
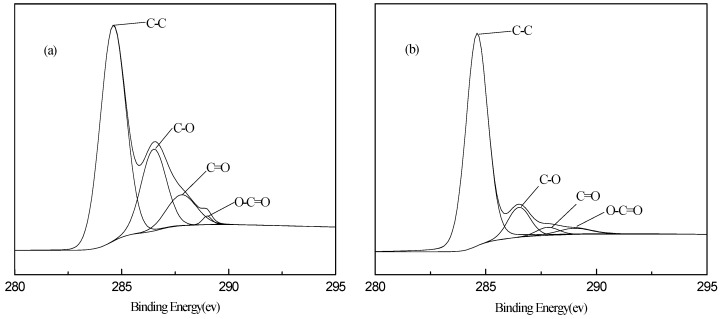
The XPS spectra of (**a**) graphite oxide and (**b**) graphene.

**Figure 7 polymers-10-01220-f007:**
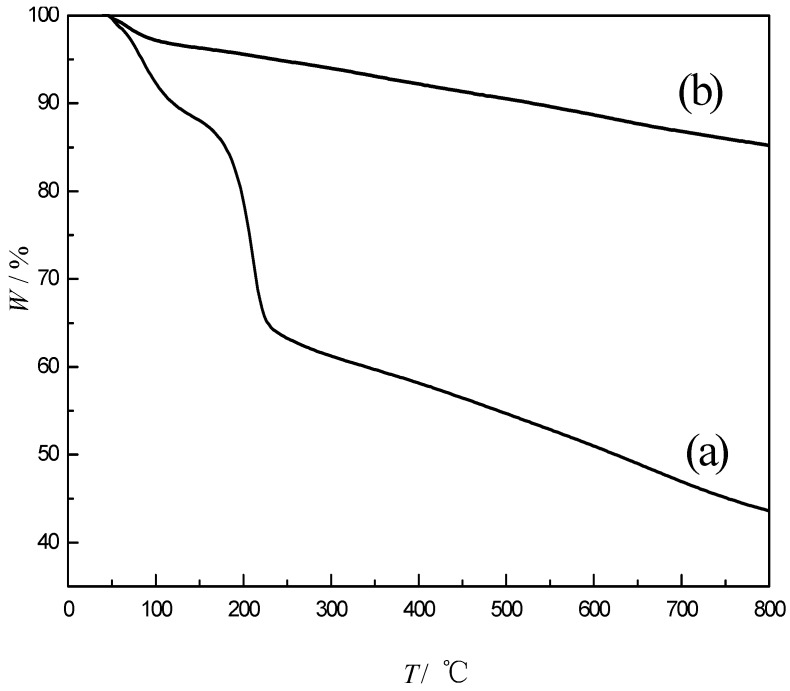
Thermogravimetric curves of (**a**) GO and (**b**) GS1.

**Figure 8 polymers-10-01220-f008:**
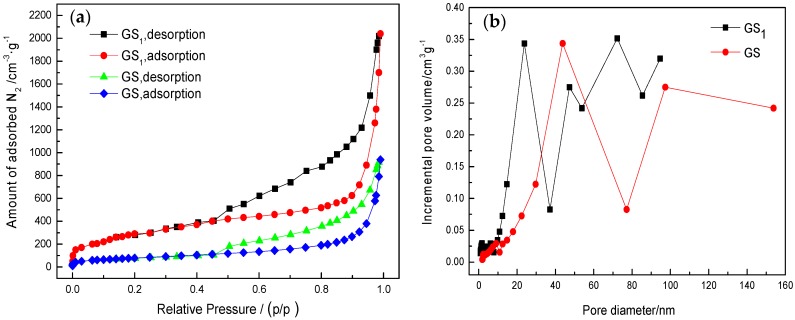
(**a**) N_2_ adsorption–desorption isotherms of GS1 and GS and (**b**) pore size distributions of GS1 and GS.

**Figure 9 polymers-10-01220-f009:**
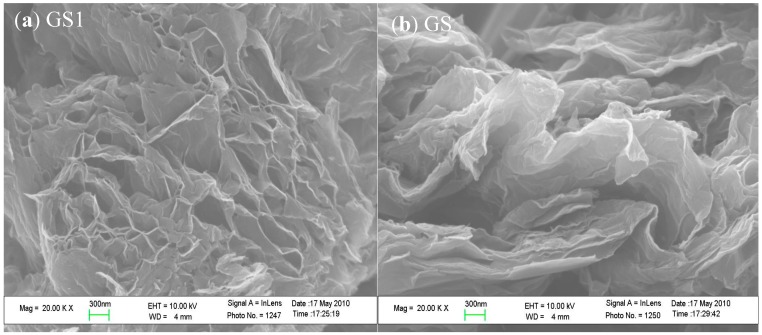
(**a**,**b**) SEM images and (**c**,**d**) TEM views of GS1 and GS.

**Figure 10 polymers-10-01220-f010:**
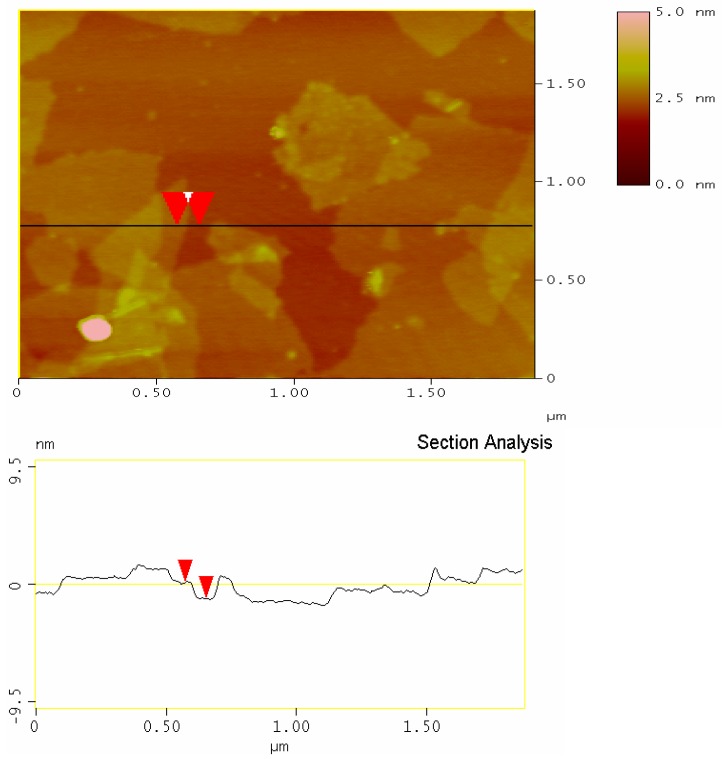
Atomic force microscopy (AFM) image of GS1.

**Figure 11 polymers-10-01220-f011:**
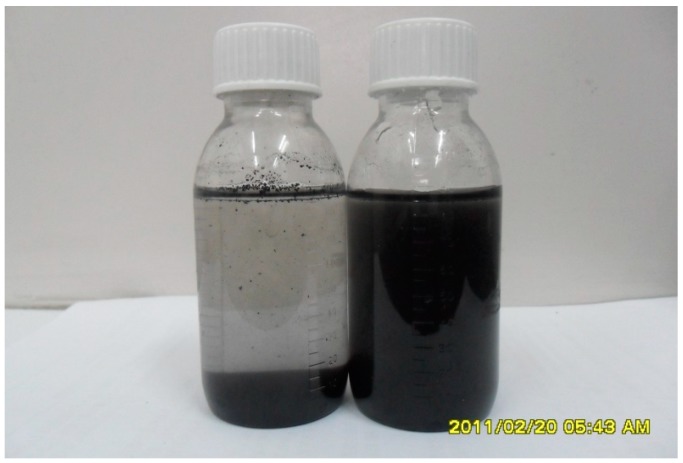
Comparison of graphene suspension before and after adding SDBS for 60 days.

**Figure 12 polymers-10-01220-f012:**
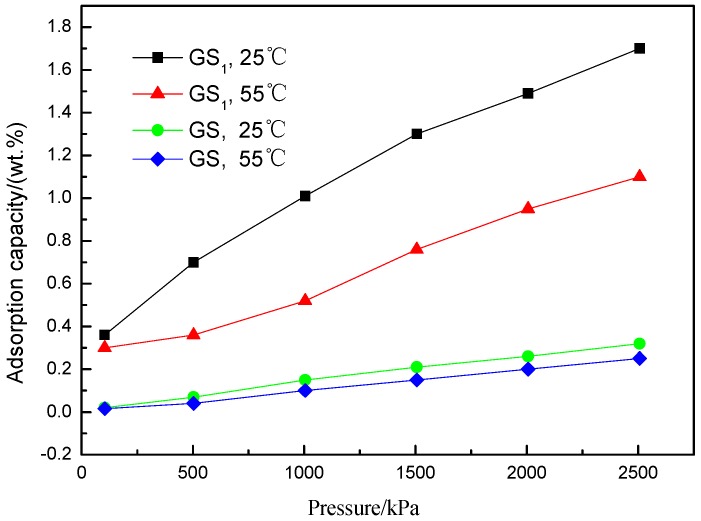
H_2_ adsorption capacity of GS1 and GS under different pressures at 25 °C and 55 °C.

**Figure 13 polymers-10-01220-f013:**
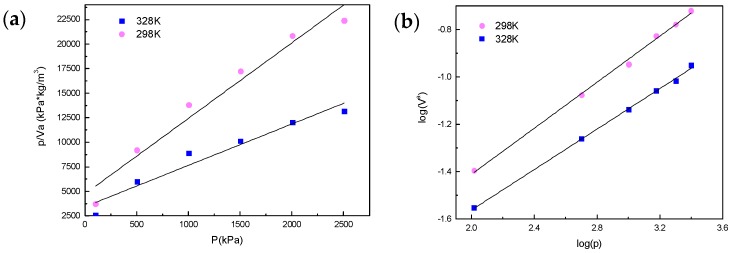
(**a**) Hydrogen adsorption capacity of graphene with the Langmuir model and (**b**) Freundlich model.

**Table 1 polymers-10-01220-t001:** Chemicals used in the experiment. SDBS: sodium dodecyl benzene sulfonate.

Name	Chemical Formula	Molecular Weight	Level	Producer
Flake Graphite	C	12.01	325 mesh, 99.8%	Alfa Aesar Tianjin Chemical Co., Ltd., Tianjin, China
Potassium Permanganate	KMnO_4_	158.04	Analytically Pure	China Sinopharm Chemical Reagent Co., Ltd., Shanghai, China
Strong Sulfuric	H_2_SO_4_	98.01	95.0–98.0%	China Sinopharm Chemical Reagent Co., Ltd., Shanghai, China
Hydrogen Peroxide	H_2_O_2_	34.01	30%	China Sinopharm Chemical Reagent Co., Ltd., Shanghai, China
Potassium Perchlorate	KClO_4_	138.56	Analytically Pure	China Sinopharm Chemical Reagent Co., Ltd., Shanghai, China
Muriatic Acid	HCl	36.56	36–38%	China Sinopharm Chemical Reagent Co., Ltd., Shanghai, China
Sodium Borohydride	NaBH_4_	37.85	Analytically Pure	China Sinopharm Chemical Reagent Co., Ltd., Shanghai, China
Sodium Nitrate	NaNO_3_	84.99	Analytically Pure	China Sinopharm Chemical Reagent Co., Ltd., Shanghai, China
Acetone	CH_3_COCH_3_	58.08	Analytically Pure	China Sinopharm Chemical Reagent Co., Ltd., Shanghai, China
SDBS	C_18_H_29_NaO_3_S	348.48	Analytically Pure	China Sinopharm Chemical Reagent Co., Ltd., Shanghai, China

**Table 2 polymers-10-01220-t002:** List of apparatus.

Equipment	Producer
Sartorius Electronic Scales	Beijing Sartorius Instrumentation System Co., Ltd., Beijing, China
DHG-9070A Electro-thermostatic Blast Oven	China Shanghai Yiheng Science and Technology Ltd., Shanghai, China
SHZ-D Circulating Vacuum Pump	China Gongyi Yuhua Instrumentation Co., Ltd., Gongyi, Henan, China
KQ-500VDF Ultrasonic Generator	China Kunshan Ultrasound Instrumentation Co., Ltd., Suzhou, Jiangsu, China
IKL RCT Basic Magnetic Stirrers	China Shanghai Yiheng Science and Technology Ltd., Shanghai, China
HC-3518 Centrifuge	China USTC Chuangxin Co., Ltd., Hefei, Anhui, China

**Table 3 polymers-10-01220-t003:** Fitting parameters of Langmuir and Freundlich adsorption isotherm equations.

*T* (K)	Langmuir Equation			Freundlich Equation		
	b (mg/g)	$V_{m}^{a}\;(\mathrm{mL}/\mathrm{mg})$	R^2^	K	n	R^2^
298	0.00123	0.2374	0.946	0.00414	0.486	0.997
328	0.00162	0.1300	0.963	0.00384	0.427	0.999

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
