# Peer review of "Preparation and Hydrogen Storage Characteristics of Surfactant-Modified Graphene"

_polymers, 2018, doi:10.3390/polym10111220_

Round 1
Reviewer 1 Report
In this manuscript, the authors proposed a new method (liquid-phase oxidation method) to prepare high quality graphene for hydrogen storage, and carried out the preparation of surfactant-modified graphene. The structure, morphology, and composition of the prepared samples were characterized by XRD, Raman, FT-IR, TG, SEM, TEM, and AFM. Besides, their hydrogen adsorption properties were investigated. This reviewer recommends accepting this manuscript after making the following revisions:
1. There are some grammatical and structure errors in the manuscript. Please correct.
2. The difference between “graphite oxide”and “graphene oxide”is not explained in the article. Please elaborate in the introduction section.
3. Some units were are written incorrectly. Please correct.
4. Some figures were not properly referred in the text. Please check throughout the manuscript and make corrections.
5. According to the description of Fig. 9, in the 3.7, “GS2” should be “GS”?
Author Response
1. There are some grammatical and structure errors in the manuscript. Please correct.
Response:Thanks for your suggestion. We have already checked those English errors carefully and corrected them..
2. The difference between “graphite oxide”and “graphene oxide”is not explained in the article. Please elaborate in the introduction section.
Response: Thanks for your suggestion. Graphene oxide can be consider as a kind of graphene, but graphite oxide is only a kind of graphite. They are acknowledged material names.
3. Some units were are written incorrectly. Please correct.
Response: Thanks for your suggestion. We have corrected those errors in the paper.
4. Some figures were not properly referred in the text. Please check throughout the manuscript and make corrections.
Response: Thanks for your suggestion. We have already checked those errors carefully and corrected them.
5. According to the description of Fig. 9, in the 3.7, “GS2” should be “GS”?
Response: Thanks for your suggestion. We have already corrected this error in the paper.

Reviewer 2 Report
The work should be published in the current form
Author Response
Response: We have already modified some errors. Thanks for your help.

Reviewer 3 Report
The manuscript entitled “Preparation and hydrogen storage characteristics of surfactant modified graphene” by Xu and coworkers investigates the use of surfactant modified graphene for energy storage applications. In this study, the modified graphene was prepared via a modified Hummers method and ultrasonic stripping from graphene oxide and NaBH4¸with the use of sodium dodecyl benzene sulfonate (SDBS) as a surfactant during the reduction process.
XRD diffraction revealed that the dispersant facilitated the dispersion of the graohene, as the graphene sample prepared with SDBS (GS1) exhibited a lower diffraction peak than that of graphene prepared in the absence of the dispersant (GS), indicating that a large distance existed between the graphene layers with GS1 existing primarily in a monolayer form.
The authors indicated that the adsorption-desorption isotherm data suggested that GS1 had a significantly larger surface area than that of the unmodified GS, albeit smaller than the theoretical surface area for a single layer of graphene. The authors attributed this to the GS1 sample existing as a combination of single and dual-layered structures. The authors note that this is improved in comparison with previous reports (277-278, but references should be provided to support this statement). It was noted that the pores of GS1 were smaller and more numerous than those of GS, which was attributed to the larger surface area of GS1 in comparison with GS. The TEM, SEM and AFM observations were also consistent with GS1 existing as fewer layers and having better dispersibility than GS, which in contrast exhibited a greater degree of stacking of layers and agglomeration. AFM analysis indicated that the GS1 sample had an average thickness of 1.1 nm, which was consistent with a single layer of graphene.
In terms of hydrogen storage and adsorption, the authors found that GS1 had a higher adsorption capacity than GS. They fitted their results to both Langmuir and Freundlich adsorption isotherms and found that both models were consistent with their experimental results. Between these two models, the best fitting was obtained with the Freundlich isotherm.
As mentioned above, the authors have employed an extensive range of characterization techniques, such as FT-IR, Raman, and XPS spectroscopy, AFM, TEM, and SEM imaging, thermogravimetric analysis, and adsorption-desorption measurements. With this experimental data in hand, I believe that this research is well-executed. This work also has practical relevance, particularly with regard to energy storage.
Overall, I believe that this manuscript is worthy of publication, pending minor revisions which are described below.
This manuscript is generally well-written, but I have included below some suggestions for possible revisions:
Line 23-24: “Hummers method” should possibly be changed to “Hummers’ method”
Line 31: “increase the increase the specific surface area” should be changed to “increase the specific surface area”.
Line 37: “researchers come to” can possibly be changed to “researchers have come to”.
Line 42: “is the curtail barrier for its wide” can be changed to “presents a barrier for its wide”, “is a current barrier impeding its wide” or “is a currently a barrier impeding its wide”.
Line 44: “highly rely on” can be changed to “rely on” or “are highly reliant on”
Line 51: “four main type” should be changed to “four main types”.
Line 63: “Hummers method” can be changed to “Hummers’ method”.
Line 89: “o-phenylenediamine and p-phenylenediamine” can be changed to “o-phenylenediamine and p-phenylenediamine” (with italics font for o- and p-).
Line 92: The phrase “reaction and sulfonate the produce in an ice” is unclear.
Line 105: “Hummers method” can be changed to “Hummers’ method”.
Figure 1, pages 3-4: Currently, Figure 1 is split between pages 3 and 4. If possible it may be best to place this figure on a single page so that the image is not split between different pages.
Line 131: “carried out by using spray drying method” can be changed to “carried out via a spray drying method” or ““carried out using a spray drying method”.
Lines 137-138: “until pH equals to 7” can possibly be changed to “until a pH of 7 was reached”.
Line 148 (Figure 2): “Graphen” should be changed to “Graphene” in the final box of Figure 2.
Line 161: “includes” can possibly be changed to “included”.
Line 165: “30 ml/min” and "28 ml/min" should be changed to “30 ml/min” and "28 ml/min", respectively (With a space between the number and units).
Line 165: “20°C-800°C” should be changed to “20-800 °C”.
Line 166: “5°C/min” should be changed to “5 °C/min” (With a space between the number and units).
Lines 177-178: References may be needed for the Brunauer-Emmett-Teller (BET) equation and the Barrett-Joyer-Halendal (BHJ) method.
Line 187: “25 °C and 55 °C” can possibly be changed to “25 and 55 °C”.
Line 199: “boarder” should be changed to “broader”.
Lines 199-200: “decreased graphene sheets gap” can possibly be changed to “decreased gap between the graphene sheets” but the existing text is also alright.
Line 203-204: “this concludes that” can possibly be changed to “this demonstrates that”, “this indicates that”, or “this suggests that”.
Line 207: “Raman characterization” can possibly be changed to “Raman spectroscopy”.
Line 208: “graphene” should be changed to “graphene”.
Line 209: A space may be needed between “(GS1).” and “The”.
Line 219: “was significantly shrunk” can possibly be changed to “had shrunken considerably”.
Line 231: “the peak of” can possibly be changed to “peaks corresponding to”.
Line 236: “have a strong peak” can be changed to “exhibit a strong peak” or “exhibit strong peaks”.
Line 243 (Figure 5 caption): “grapheme” should be changed to “graphene”.
Line 246: “283eV and 534eV” should be changed to “283 and 534 eV”.
Line 250: “285eV and 287eV” should be changed to “285 and 287 eV”.
Line 251: “289eV” should be changed to “289 eV”.
Lines 251-252: The phrase “After reduction, the characteristic peaks of graphene, including epoxy and carboxyl groups are” is a little unclear and may need clarification. Maybe this can be changed to “After reduction, the characteristic peaks of the modified graphene, including epoxy and carboxyl groups are” or “After reduction, the characteristic peaks of the surfactant-modified graphene, including epoxy and carboxyl groups are”.
Line 263: “that the graphene sample” should possibly be changed to “that the GS1 sample”.
Lines 277-278: “However, compared with previous studies, the specific surface is still significantly improved”. The authors should provide a reference for this statement.
Line 282: “photographs” can possibly be changed to “images”.
Line 288: “fewer defects while GS1 contains fewer” can be changed to “fewer defects while also containing fewer”.
Line 311: “sinks into the bottom” can be changed to “sinks to the bottom”.
Line 312: “As the dispersant” can possibly be changed to “The dispersant”.
Line 320: “25 °C and 55 °C” can be changed to “25 and 55 °C”.
Line 325: The phrase “in the micropores with a small amount” is a little unclear.
Lines 330-332: The authors may need to provide references for the Clausius-Clapeyron Equation and for the recommended adsorption heat of 15 kJ/mol which is attributed to the United States Department of Energy.
Line 362: “Form the table 3” can be changed to “From Table 3”.
Line 362: “298K and 328K” can be changed to “298 and 328 K”.
Lines 375-376: “by adding dispersant sodium dodecyl benzene sulfonate (SDBS) during” can be changed to “by adding the dispersant sodium dodecyl benzene sulfonate (SDBS) during” or “by adding sodium dodecyl benzene sulfonate (SDBS) as a dispersant during”.
Line 383: “with thickness of 1.1 nm and small amount of” can be changed to “with a thickness of 1.1 nm and a small amount of”.
Figure name format in the text: The format of the Figure names appearing in this manuscript seem to vary, with formats such as Figure.1, Fig. 7, or Figure 10. I believe that for this journal the format with the full name rather than the abbreviations is required (for example, Figure 10), but the authors may need to check the MDPI guidelines to confirm this. Lines 106, 130, 278, 283, 284, 287, 288.
Author Response
Line 23-24: “Hummers method” should possibly be changed to “Hummers’ method”
Response: We have modified this expression as “Hummers’ method”.
Line 31: “increase the increase the specific surface area” should be changed to “increase the specific surface area”.
Response: We have revised this expression as “increase the specific surface area”.
Line 37: “researchers come to” can possibly be changed to “researchers have come to”.
Response: We have revised this expression as “researchers have come to”.
Line 42: “is the curtail barrier for its wide” can be changed to “presents a barrier for its wide”, “is a current barrier impeding its wide” or “is a currently a barrier impeding its wide”.
Response: We have corrected this as “is a crucial barrier impeding its wide application”.
Line 44: “highly rely on” can be changed to “rely on” or “are highly reliant on”
Response: “highly rely on” has been changed to “rely on”.
Line 51: “four main type” should be changed to “four main types”.
Response: “four main type” is changed to “four main types”.
Line 63: “Hummers method” can be changed to “Hummers’ method”.
Response: “Hummers method” is changed to “Hummers’ method”.
Line 89: “o-phenylenediamine and p-phenylenediamine” can be changed to “o-phenylenediamine and p-phenylenediamine” (with italics font for o- and p-).
Response:Italics font for o- and p- is adopted.
Line 92: The phrase “reaction and sulfonate the produce in an ice” is unclear.
Response:We have corrected this phrase, as “and then the product was sulfonated in an ice bath for 2 hours”.
Line 105: “Hummers method” can be changed to “Hummers’ method”.
Response: “Hummers method” is changed to “Hummers’ method”.
Figure 1, pages 3-4: Currently, Figure 1 is split between pages 3 and 4. If possible it may be best to place this figure on a single page so that the image is not split between different pages.
Response: We have placed Figure 1 on a single page.
Line 131: “carried out by using spray drying method” can be changed to “carried out via a spray drying method” or ““carried out using a spray drying method”.
Response: This phrase is changed to “carried out via a spray drying method”.
Lines 137-138: “until pH equals to 7” can possibly be changed to “until a pH of 7 was reached”.
Response: “until a pH of 7 was reached” has been used.
Line 148 (Figure 2): “Graphen” should be changed to “Graphene” in the final box of Figure 2.
Response: “Graphene” in the final box of Figure 2 is corrected.
Line 161: “includes” can possibly be changed to “included”.
Response: “included” is used.
Line 165: “30 ml/min” and "28 ml/min" should be changed to “30 ml/min” and "28 ml/min", respectively (With a space between the number and units).
Response: A space between the number and units has been added.
Line 165: “20°C-800°C” should be changed to “20-800 °C”.
Response: “20-800 °C” is used.
Line 166: “5°C/min” should be changed to “5 °C/min” (With a space between the number and units).
Response: A space is added.
Lines 177-178: References may be needed for the Brunauer-Emmett-Teller (BET) equation and the Barrett-Joyer-Halendal (BHJ) method.
Response: We have added two references on BET equation and BHJ method in the paper.
Line 187: “25 °C and 55 °C” can possibly be changed to “25 and 55 °C”.
Response: “25 and 55 °C” is used.
Line 199: “boarder” should be changed to “broader”.
Response: “boarder” is changed to “broader”.
Lines 199-200: “decreased graphene sheets gap” can possibly be changed to “decreased gap between the graphene sheets” but the existing text is also alright.
Response: “decreased gap between the graphene sheets” is used.
Line 203-204: “this concludes that” can possibly be changed to “this demonstrates that”, “this indicates that”, or “this suggests that”.
Response: “This demonstrates that” is used.
Line 207: “Raman characterization” can possibly be changed to “Raman spectroscopy”.
Response: “Raman spectroscopy” is used.
Line 208: “grapheme” should be changed to “graphene”.
Response: “grapheme” is changed to “graphene”.
Line 209: A space may be needed between “(GS1).” and “The”.
Response: A space between “(GS1).” and “The” has been added.
Line 219: “was significantly shrunk” can possibly be changed to “had shrunken considerably”.
Response: “had shrunken considerably” is used.
Line 231: “the peak of” can possibly be changed to “peaks corresponding to”.
Response: “peaks corresponding to” is used.
Line 236: “have a strong peak” can be changed to “exhibit a strong peak” or “exhibit strong peaks”.
Response: “exhibit strong peaks” is used.
Line 243 (Figure 5 caption): “grapheme” should be changed to “graphene”.
Response: “grapheme” is changed to “graphene”.
Line 246: “283eV and 534eV” should be changed to “283 and 534 eV”.
Response: “283 and 534 eV” is used.
Line 250: “285eV and 287eV” should be changed to “285 and 287 eV”.
Response: “285eV and 287eV” is changed to “285 and 287 eV”.
Line 251: “289eV” should be changed to “289 eV”.
Response: A space is added.
Lines 251-252: The phrase “After reduction, the characteristic peaks of graphene, including epoxy and carboxyl groups are” is a little unclear and may need clarification. Maybe this can be changed to “After reduction, the characteristic peaks of the modified graphene, including epoxy and carboxyl groups are” or “After reduction, the characteristic peaks of the surfactant-modified graphene, including epoxy and carboxyl groups are”.
Response: “After reduction, the characteristic peaks of the modified graphene, including epoxy and carboxyl groups are” is used.
Line 263: “that the graphene sample” should possibly be changed to “that the GS1 sample”.
Response: “that the GS1 sample” is used.
Lines 277-278: “However, compared with previous studies, the specific surface is still significantly improved”. The authors should provide a reference for this statement.
Response: Thanks for your good suggestion. We have already added Reference [27] in the paper.
Line 282: “photographs” can possibly be changed to “images”.
Response: “photographs” is changed to “images”.
Line 288: “fewer defects while GS1 contains fewer” can be changed to “fewer defects while also containing fewer”.
Response: “fewer defects while GS1 contains fewer” is changed to “fewer defects while also containing fewer”.
Line 311: “sinks into the bottom” can be changed to “sinks to the bottom”.
Response: “sinks into the bottom” is changed to “sinks to the bottom”.
Line 312: “As the dispersant” can possibly be changed to “The dispersant”.
Response: “As the dispersant” is changed to “The dispersant”.
Line 320: “25 °C and 55 °C” can be changed to “25 and 55 °C”.
Response: “25 °C and 55 °C” is changed to “25 and 55 °C”.
Line 325: The phrase “in the micropores with a small amount” is a little unclear.
Response: We have corrected this expression as “hydrogen molecules are mainly adsorbed in the micropores”.
Lines 330-332: The authors may need to provide references for the Clausius-Clapeyron Equation and for the recommended adsorption heat of 15 kJ/mol which is attributed to the United States Department of Energy.
Response: Thanks for your good suggestion. We have already added two references in this paper.
Line 362: “Form the table 3” can be changed to “From Table 3”.
Response: “Form the table 3” is changed to “From Table 3”.
Line 362: “298K and 328K” can be changed to “298 and 328 K”.
Response: “298K and 328K” is changed to “298 and 328 K”.
Lines 375-376: “by adding dispersant sodium dodecyl benzene sulfonate (SDBS) during” can be changed to “by adding the dispersant sodium dodecyl benzene sulfonate (SDBS) during” or “by adding sodium dodecyl benzene sulfonate (SDBS) as a dispersant during”.
Response: “by adding the dispersant sodium dodecyl benzene sulfonate (SDBS) during” is used.
Line 383: “with thickness of 1.1 nm and small amount of” can be changed to “with a thickness of 1.1 nm and a small amount of”.
Response: “with thickness of 1.1 nm and small amount of” is changed to “with a thickness of 1.1 nm and a small amount of”.
Figure name format in the text: The format of the Figure names appearing in this manuscript seem to vary, with formats such as Figure.1, Fig. 7, or Figure 10. I believe that for this journal the format with the full name rather than the abbreviations is required (for example, Figure 10), but the authors may need to check the MDPI guidelines to confirm this. Lines 106, 130, 278, 283, 284, 287, 288.
Response: All the figure captions have been corrected as “Figure 1, …, Figure 13”.
